biomaterials

titanium implant, nanostructure, calcium ions, osseointegration, hydrothermal treatment

**Author for correspondence:**
Lei Zhang
e-mail: zhangleistone@sina.com

†These authors equally contributed to this study.

This article has been edited by the Royal Society of Chemistry, including the commissioning, peer review process and editorial aspects up to the point of acceptance.

# Synergistic effect of nanostructure and calcium ions on improving the bioactivity of titanium implants

Yue Zhang[1,†], Jingwen Wang[1,†],
Shahrzad Hosseinijenab[1], Yiqiang Yu[1], Chao Lv[1],
Cheng Luo[1], Weijie Zhang[1], Xi Sun[2] and Lei Zhang[1]

[1]Department of Prosthodontics, and [2]Department of Endodontics, School and Hospital of Stomatology, Tongji University, Shanghai Engineering Research Center of Tooth Restoration and Regeneration, Shanghai 200072, People's Republic of China

LZ, 0000-0002-4548-338X

Surface structure and composition play essential roles in the osseointegration of titanium implants. In the present study, a nanoscale surface structure incorporated with calcium ions was fabricated on a titanium surface by hydrothermal treatment. The characteristics of the surfaces were analysed, and the bioactivity of the samples was evaluated *in vitro* and *in vivo*. nm-Ti and nm/Ca-Ti surfaces were significantly more hydrophilic than control-Ti surfaces. nm/Ca-Ti samples showed much faster bone-like apatite precipitation in simulated body fluid than the other samples. The results of MC3T3-E1 cell tests demonstrated that both nm-Ti and nm/Ca-Ti surfaces accelerated cell adhesion and proliferation. The highest level of osteogenesis-related genes (Runx2, bone morphogenetic protein-2, osteopontin and osteocalcin) were observed in nm/Ca-Ti samples, followed by nm-Ti samples. Alizarin red staining experiment showed that the amount of extracellular matrix mineralized nodules in nm/Ca-Ti group was significantly higher than others. In animal experiments using SD rats, nm/Ca-Ti showed the highest value of new bone formation at two and four weeks. The present study suggested that the nanostructure and calcium ions showed synergetic effects on accelerating bone-like apatite precipitation and osteoblast cell growth and differentiation. Animal experiment further indicated that such surface could promote early osteogenesis.

# 1. Introduction

Pure titanium and titanium alloys are widely used for manufacturing dental implants due to their excellent chemical and mechanical properties [1]. Despite the great clinical success, a healing period of three months is still inevitable for achieving stable osseointegration after surgery, which limits the application of immediate or early loading techniques especially in patients with compromised osteogenic capacity [2]. To shorten the bone formation time, many studies have been performed to improve the bioactivity of implants by changing their surface characteristics.

One of the most popular methods is coating with hydroxyapatite, which can significantly improve the cellular attachment and proliferation of osteoblasts [3–5]. However, adverse effects have also been reported during long-term application, such as delamination of the coating layer, causing mechanical failure, dissociation and release of some ions, and leading to alveolar bone resorption [6,7]. Some researchers reported the incorporation of bioactive ions, such as strontium, magnesium and calcium ions, into the surface of titanium [8,9]. These ions accelerate osteogenic formation and inhibit the activity of osteoclasts. Among these ions, calcium, one of the essential elements in hard tissue, has recieved more attention. Studies have revealed that calcium can contribute to the differentiation of mesenchymal cells into osteoblastic cells [10,11]. In addition, calcium was reported to efficiently decrease the incidence of infection, especially infections caused by methicillin-resistant *Staphylococcus aureus* [12]. One of our previous studies showed that Ca-bound Ti surfaces can be fabricated through hydrothermal treatment. This kind of Ti surface exhibited good bioconductivity [13].

Good hydrophilicity plays an important role in cell adherence and proliferation. Zuldesmi *et al*. [14] reported that the surface wettability of implants was a crucial factor in their osteoconductivity. Implants modified after hydrothermal treatment possess fantastic surface hydrophilicity, and their bioactivity is effectively improved [15].

Nanostructure benefit new bone formation, so many researchers have struggled to form surfaces with nanostructured topography [16–18]. As reported, nanostructures are effective at promoting the maturation of focal adhesions and the formation of filopodia [19]. As a result, nanostructure enhances cell attachment, proliferation, osteogenesis gene expression and mineralization [20]. This structure may promote early osseointegration. For example, anodic oxidation and microarc oxidation are used to produce nanotubes on the surface of implants and the capacity of bone formation is significantly improved [15,21].

Therefore, in this study, we tried to construct a nanostructured titanium surface incorporated with calcium ions under the hypothesis that these two factors may show some synergistic effect in promoting the osteogenic activity of titanium implants. The physical and chemical characteristics of the treated surfaces were analysed. The bioactivity of the samples was investigated *in vitro* and further confirmed by a short-term *in vivo* study.

# 2. Material and methods

## 2.1. Specimen preparation

Commercially pure Ti samples ($10 \times 10 \times 2$ mm$^3$ plates) with machined surface were used. The pure titanium rods with a diameter of 2 mm and a length of 4 mm are used as titanium implants for animal experiment. All samples were ultrasonically cleaned in acetone, ethanol and distilled water, and divided into three groups. The control-Ti group consisted of untreated samples. In the nm-Ti group, samples were hydrothermally treated in 0.5 M sodium hydroxide solutions at 180°C for 24 h. In the nm/Ca-Ti group, samples were prepared by a two-step procedure. In the first step, they were hydrothermally treated in 0.5 M sodium hydroxide solutions at 180°C for 24 h, followed by the second step: hydrothermal treatment in 2 mM calcium oxide solutions at 180°C for 24 h.

## 2.2. Surface characteristics

The surface morphology of the control-Ti, nm-Ti and nm/Ca-Ti samples was observed by using scanning electron microscope (SEM, S-4800; Hitachi, Chiyoda, Japan). The chemical compositions of the surface layer of different samples were analysed by X-ray photoelectron spectroscopy (XPS, Quantera SXM, ULVAC-PHI, Tokyo, Japan) and thin-film X-ray diffractometry (XRD, X'Pert-APD, Philips, Almelo, The Netherlands). The surface hydrophilicity of different samples was evaluated by measuring the water contact angle with a Dataphysics instrument (OCA25).

**Table 1.** Composition of SBF solution.

| ions | Na$^+$ | K$^+$ | Mg$^{2+}$ | Ca$^{2+}$ | Cl$^{2-}$ | CO$_3{}^{2-}$ | HPO$_4{}^{2-}$ | SO$_4{}^{2-}$ |
|---|---|---|---|---|---|---|---|---|
| concentration (mM) | 213 | 7.5 | 2.25 | 3.75 | 221.7 | 3.75 | 1.5 | 0.5 |

## 2.3. Initial bioactivity assessment in simulated body fluid

Samples were immersed in simulated body fluid (SBF) for 3 and 7 days for initial evaluation of bioactivity *in vitro*. The composition of the SBF solution is listed in table 1, and the pH was adjusted to 7.4 by adding HCl [22]. Each disc was immersed in 15 ml SBF for 3 and 7 days at 37°C with shaking. The SBF solutions were replaced every day. After immersion, the discs were gently washed using deionized water and dried in air. Samples after immersion in SBF were observed under SEM and analysed by XRD.

## 2.4. Osteoblast culture

MC3T3-E1 cells were used in this study to evaluate cell attachment, proliferation and differentiation. Osteoblasts were cultured with α-MEM medium with 10% fetal bovine serum and 5% penicillin-streptomycin at 37°C, 5% CO$_2$. After confluence, cells were seeded onto specimens in 24-well plates.

### 2.4.1. Cell morphology observation

MC3T3-E1 cells were seeded on the surface of different samples at a density of $2 \times 10^4$ per well and cultured at 37°C, 5% CO$_2$ for 6 and 24 h. Samples were gently washed with phosphate buffer saline (PBS) three times and then fixed at room temperature for 2 h with 2.5% glutaraldehyde. After fixation, the specimens were gently washed with PBS and dehydrated in an ascending series of ethanol concentrations (50%, 60%, 70%, 80%, 90%, 100% and 100%). Then, the specimens were immersed in tert-butanol three times. After freezing for 1 h at −20°C, the samples were dried with a freeze dryer and then coated with gold, and the morphology of MC3T3-E1 cells were observed by SEM.

### 2.4.2. Cells proliferation measurement

MC3T3-E1 cells were seeded onto the plates of different groups placed at a density of $1 \times 10^4$ cells per well. After 1, 4 and 7 days of cultivation, the cell number on each sample was evaluated using the Cell Counting Kit-8 assay (CCK-8, Beyotime). At the appointed time, specimens were gently washed with PBS and placed into a new culture plate. A total of 550 µl CCK-8 solution (1 : 10 dilution with culture medium) was added to each well and samples were incubated at 37°C for 3 h. Then, the OD value of each solution was measured using a microplate reader at 450 nm.

## 2.5. Real-time reverse transcription polymerase chain reaction

Relative mRNA levels of osteogenesis genes were analysed through quantitative reverse transcription polymerase chain reaction (qRT-PCR). The cells were seeded at a density of $2 \times 10^4$ cells per well on specimens in 24-well culture plates and cultured with osteogenic inducer (Sigma, 0.1 µM dexamethasone, 50 µM L-ascorbic acid and 10 mM β-glycerophosphate) for 7 and 10 days. At each time point, total RNA was extracted using TRIzol reagent (Invitrogen), and reverse transcription was performed using the obtained RNA to create complementary DNA (cDNA) with the PrimeScript RT reagent Kit (Roche). Then, real-time PCR were performed using SYBR Premix Ex Taq II (Roche). Four typical osteogenesis-related genes were measured: osteopontin (OPN), osteocalcin (OCN), bone morphogenetic protein-2 (BMP-2) and Runx2. The quantities of the target genes were normalized by using the housekeeping gene GAPHD as the internal control gene.

## 2.6. Alizarin red staining experiment

*In vitro* mineralization of MC3T3-E1 cells was observed through Alizarin red staining experiment. The cells were seeded at a density of $1 \times 10^4$ cells per well on specimens in 24-well culture plates. After the cells adhesion for 24 h, the ordinary culture solution was replaced with osteogenic inducer (Sigma,

0.1 µM dexamethasone, 50 µM L-ascorbic acid and 10 mM β-glycerophosphate). The cells were cultured for 21 days, and the culture medium was changed every 3 days.

Samples were gently washed with PBS three times and then stained at room temperature for 5 min with Alizarin red dye 1 ml each well. Samples were cleaned by soaking and washing with PBS several times until the liquid is no longer discoloured. The mineralized nodules were observed by stereomicroscope and photographed at different magnifications after the samples dried naturally.

Spectrophotometric determination of mineralized nodules with Alizarin red and cetyl pyridinium chloride was performed. Mineralized nodules were eluted by adding 1 ml 10% cetyl pyridinium chloride into each well and shaking at 37°C for 15 min. Then, the OD value of each solution was measured at 562 nm.

## 2.7. Animal experiment

Two-month-old female SD rats were used for animal experiment. Titanium rods of control-Ti, nm-Ti and nm/Ca-Ti were implanted into bilateral femur of the rats, respectively. A round bur with a diameter of 1.9 mm was used to prepare the implant hole with a depth of 3 mm. After the Ti cylinder was slightly pushed into the hole, the muscle and skin were sutured tightly layer by layer. Three groups of titanium rods were randomly implanted into the femurs of rats.

## 2.8. Micro-CT evaluation

Specimens were retrieved and fixed in 4% paraformaldehyde at 4°C for 48 h. The specimens were examined by a micro-CT scanner (µCT50, Scanco Medical, Switzerland). The micro-CT images were reconstructed by Evaluation Program v. 6.6. The bone volumes (BV) within 100 µm around the implant were measured as trabecular bone volume fraction (BV/TV). And other quantitative analysis of bone around implants was made, including trabecular number (Tb.N), trabecular thickness (Tb.Th) and trabecular separation (Tb.Sp).

## 2.9. Histological observation

After micro-CT evaluation, the specimens were placed in the dehydrator (Exakt300, German) for gradient dehydration treatment with 70%, 75%, 80%, 85%, 90%, 95%, 100% and 100% ethanol solution. Then the samples were embedded in methyl methacrylate (MMA) resin and set on the microtome (Exakt300, Germany) for cutting. Finally, the slices were stained with Van Gieson (VG) for histological observation.

## 2.10. Statistical analysis

All of the experiments were repeated three times separately. The data were collected and analysed using IBM SPSS v. 22.0.0 Data. One-way ANOVA combined with Student–Newman–Keuls *post hoc* tests were used to determine the level of significance, and $p < 0.05$ was considered to be significant.

# 3. Results

## 3.1. Analysis of surface characteristics

SEM images of the sample surfaces are shown in figure 1. The control-Ti samples showed a smooth surface (figure 1*a,d*). On nm-Ti and nm/Ca-Ti, a similar porous structure could be observed (figure 1*b,c*). Under higher magnification, it was very clear that the porous surface was composed of sheets with a thickness of $27 \pm 2$ nm (figure 1*e,f*).

The results of the XPS analysis of the specimens are shown in table 2. Except for Ti, O and C, trace amounts of Ca and Na existed on the surface of control-Ti. While on the nm-Ti, a high Na content can be detected. A high amount of Ca can be seen on the surface of nm/Ca-Ti specimens.

The XRD patterns of different titanium specimens are shown in figure 2. Ti peaks were detected on the surface of control-Ti (figure 2*a*). The $Na_2Ti_6O_{13}$ peak can be found on the surface of nm-Ti (figure 2*b*). The $CaTiO_3$ peak was detected on the nm/Ca-Ti surface (figure 2*c*).

The surface hydrophilicity of different titanium implants was estimated by the water contact angles. As shown in figure 3, there was a significant difference between control-Ti, nm-Ti and nm/Ca-T. The

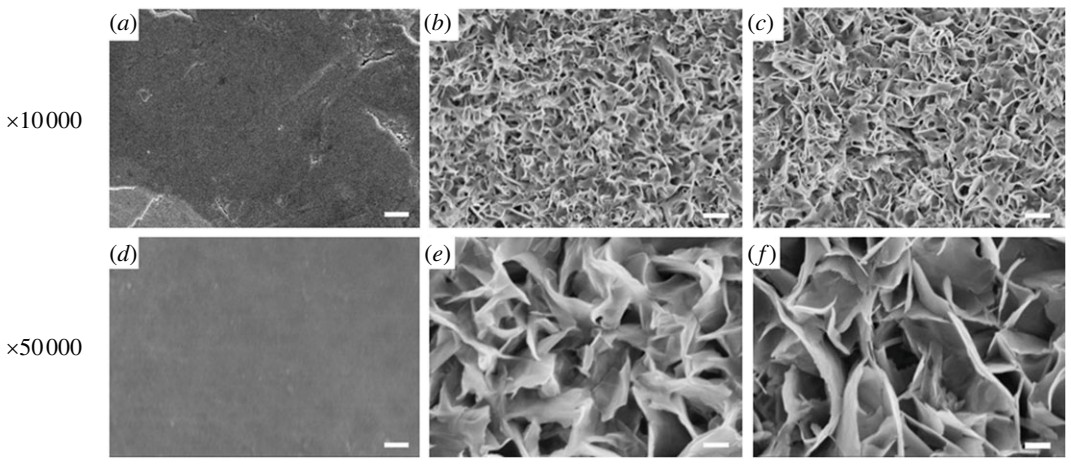

**Figure 1.** SEM images of different substrates: (*a,d*) control-Ti, (*b,e*) nm-Ti and (*c,f*) nm/Ca-Ti. (Scale bar of (*a–c*) = 10 μm, scale bar of (*d–f*) = 2 μm).

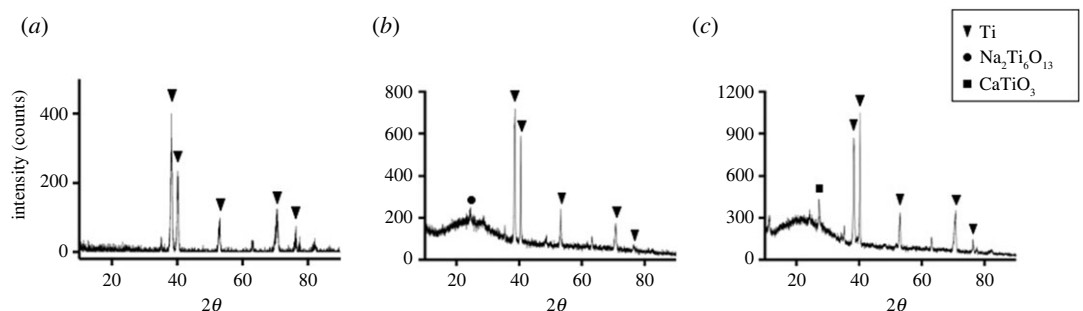

**Figure 2.** X-ray diffraction patterns of the Ti specimens: (*a*) control-Ti, (*b*) nm-Ti and (*c*) nm/Ca-Ti.

**Table 2.** Element content on the surface of samples determined by XPS.

|  | C (%) | O (%) | Ti (%) | Ca (%) | Na (%) |
|---|---|---|---|---|---|
| control-Ti | 50.11 | 42.58 | 6.79 | 0.22 | 0.30 |
| nm-Ti | 27.88 | 49.47 | 16.79 | 0.14 | 5.11 |
| nm/Ca-Ti | 25.55 | 52.01 | 14.30 | 7.21 | 0.20 |

surface of nm-Ti with an average contact angle of 52.41° showed significantly higher hydrophilicity than other surfaces, and the second wettability surface was nm/Ca-Ti (97.14°). The average contact angle of the control-Ti was 118.32°, showing the lowest hydrophilicity.

## 3.2. Characterization of the surface after immersion in SBF

Figure 4 shows SEM images of the control-Ti, nm-Ti and nm/Ca-Ti surfaces after immersion in SBF for 3 and 7 days. As shown, there was no apatite deposition on the surface of control-Ti (figure 4*a*). A small amount of apatite was observed on nm-Ti specimens both at days 3 and 7 (figure 4*b*). Nevertheless, the whole surface of nm/Ca-Ti was covered with the deposition both at days 3 and 7 after soaking in SBF (figure 4*c*). However, hydroxyapatite was detected only on the nm/Ca-Ti samples at day 7 by XRD (figure 4*d–f*).

## 3.3. Cells adhesion and spreading

The morphology of MC3T3-E1 cells on different specimens was observed by SEM. After 6 h of cultivation, cells on control-Ti spread poorly and showed a spindle shape (figure 5*a*). On the other

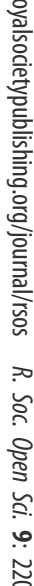

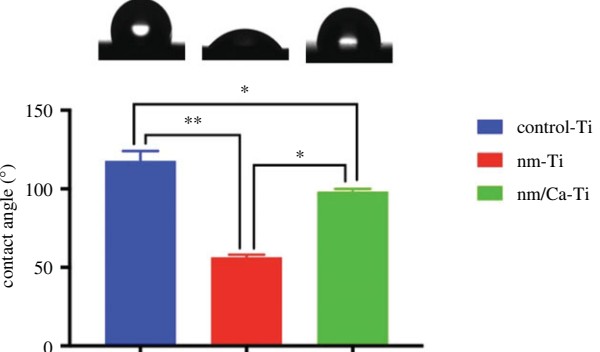

**Figure 3.** Water contact angle of different surfaces: control-Ti, nm-Ti and nm/Ca-Ti surface. ($^{*}p < 0.05$, $^{**}p < 0.01$).

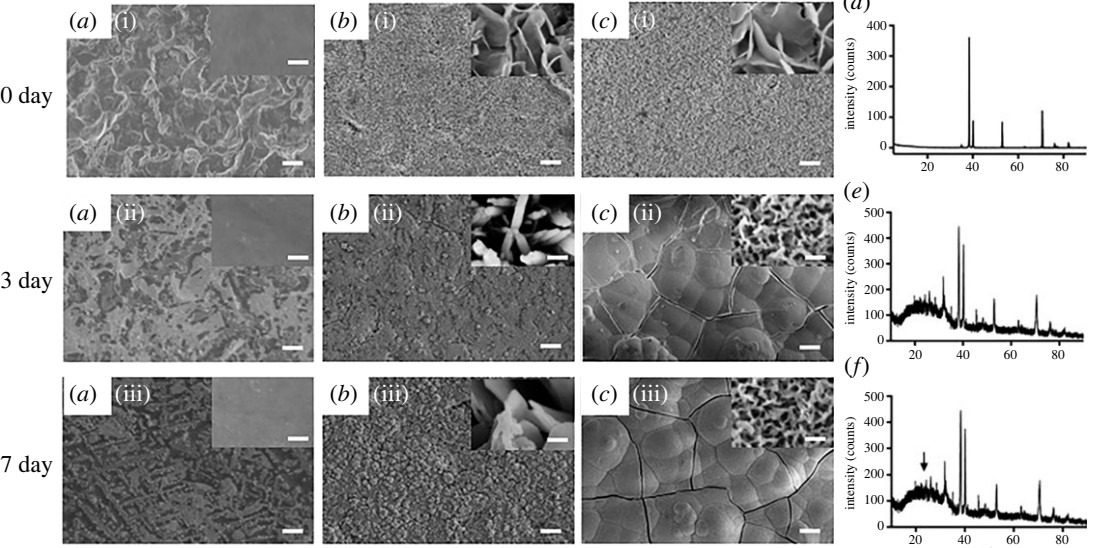

**Figure 4.** (a–c) SEM images of different samples after soaking in SBF for 3 and 7 days. (Scale bar of low magnification = 10 μm, with a magnification of 1000, scale bar of high magnification = 200 nm, with a magnification of 50 000). (d–f) XRD patterns of the specimens at day 7, (d) control-Ti, (e) nm-Ti and (f) nm/Ca-Ti. (the arrow marks the hydroxyapatite).

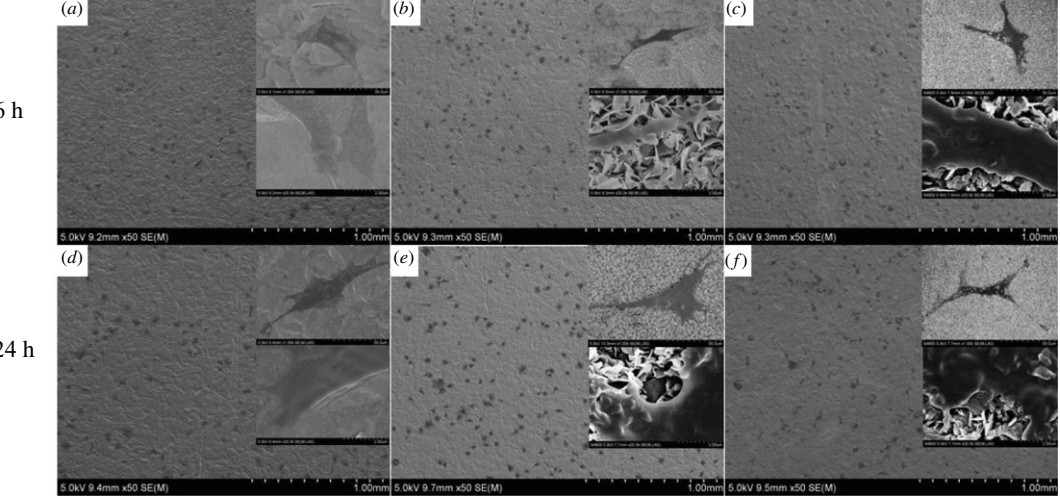

**Figure 5.** (a–f) SEM images of MC3T3-E1 on different samples at 6 h and 24 h.

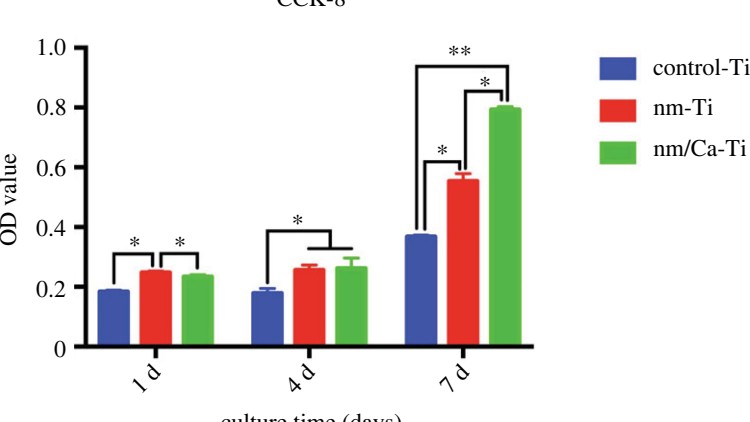

**Figure 6.** Proliferation of MC3T3-E1 cells cultured on the control-Ti, nm-Ti and nm/Ca-Ti surfaces after incubation for 1, 4 and 7 days. ($^*p < 0.05$, $^{**}p < 0.01$).

hand, cells on nm-Ti and nm/Ca-Ti exhibited a typical polygonal shape (figure 5b,c). Moreover, compared with those on the control-Ti, these cells showed more pseudopods. With prolonged incubation time, a similar tendency could be confirmed. As shown in figure 5d–f, the cell number on all samples increased dramatically after 24 h, and the nm-Ti samples showed the highest cell density.

## 3.4. Cells proliferation

Figure 6 shows MC3T3-E1 cell proliferation on different samples. The number of cells increased significantly from 1 to 7 days in all groups. The nm-Ti surface displayed the highest absorbance value at day 1. However, the cell number increased faster on the nm/Ca-Ti surface, and there was no significant difference between nm/Ca-Ti and nm-Ti at day 4. At day 7, the highest OD value was observed in the nm/Ca-Ti group ($p < 0.05$).

## 3.5. Osteogenesis-related genes expression

After 7 and 10 days' culture, osteogenesis-related gene expression in MC3T3-E1 cells on different samples was evaluated. The expression levels of Runx2, BMP-2, OPN and OCN are shown in figure 7. Runx2 mRNA expression peaked at 7 days in all groups. Compared with those on the control-Ti, the expression levels on the nm-Ti and nm/Ca-Ti surfaces were significantly higher at 7 and 10 days of culture ($p < 0.05$ or $p < 0.01$). MC3T3-E1 cells grown on the nm/Ca-Ti surface displayed markedly greater levels of BMP-2 mRNA expression than the control-Ti and nm-Ti surface at 7 days ($p < 0.05$). At 10 days, the expression levels of BMP-2 on nm-Ti and nm/Ca-Ti samples were significantly higher than those on control-Ti. Moreover, the mRNA expression of OCN at both 7 and 10 days was markedly higher on the surface of nm-Ti and nm/Ca-Ti than on the surface of control-Ti. The pattern of OPN mRNA expression was similar to that of OCN expression. nm-Ti and nm/Ca-Ti showed significantly higher OPN levels at both day 7 and day 10.

## 3.6. Alizarin red staining experiment

Alizarin red staining showed that mineralized nodules were formed on the surface of titanium discs in each group after 21 days of culture (figure 8a). A few red mineralized nodules were seen on the surface of control-Ti group. There were more and thicker nodules on the surface of nm-Ti group. And in nm/Ca-Ti group, the surface of samples was almost covered by mineralized nodules with red background. As shown in figure 8b, the quantitative measurement results showed that the amount of extracellular matrix (ECM) mineralized nodules on the surface in nm/Ca-Ti group was significantly higher than that in the other two groups. And there was no significant difference between nm-Ti group and control-Ti group.

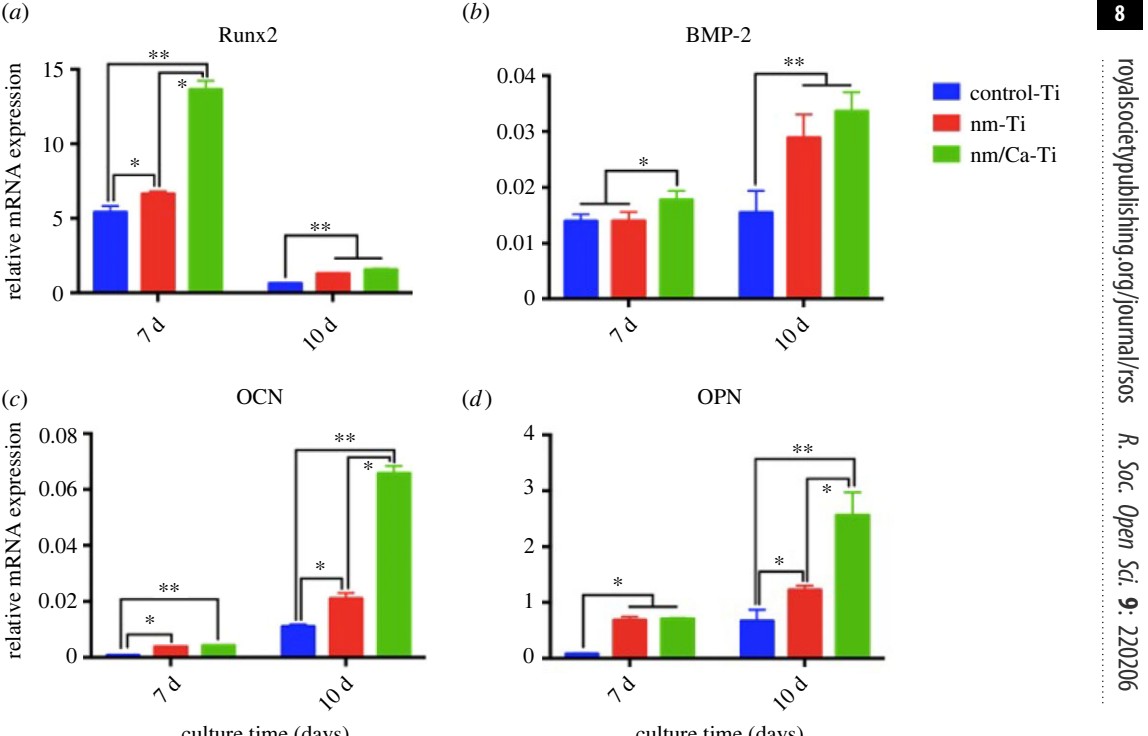

**Figure 7.** Gene expression of MC3T3-E1 cultured on the control-Ti, nm-Ti and nm/Ca-Ti surfaces after incubation for 7 and 10 days: (*a*) Runx2, (*b*) BMP-2, (*c*) OCN and (*d*) OPN. ($^* p < 0.05$, $^{**} p < 0.01$).

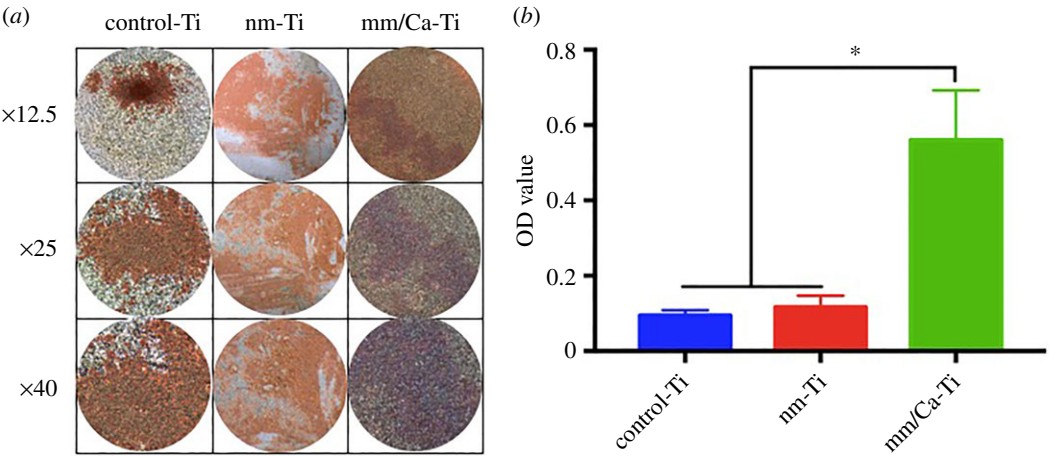

**Figure 8.** Results of Alizarin red staining after 21 days culture: (*a*) Photos at different magnification after alizarin red staining of different groups. (*b*) Quantitative analysis of mineralized nodules on the surfaces of titanium sheets in each group after elution. ($^* p < 0.05$, $^{**} p < 0.01$).

## 3.7. Animal experiment

The bone–implant interface was analysed by VG staining. The results are shown in figure 9. At two weeks, less new bone tissue was formed around control-Ti group. In control-Ti and nm-Ti groups, an obvious distance between the bone tissue and the implant could be observed, showing a poor osseointegration. As for nm/Ca-Ti group, new bone was in close contact with the surface of the titanium implant. At four weeks, the surfaces with direct bone contact of all the three groups increased. But in some areas of control-Ti and nm-Ti surfaces, there was still a gap between the implant and the bone.

control-Ti          nm-Ti          mm/Ca-Ti

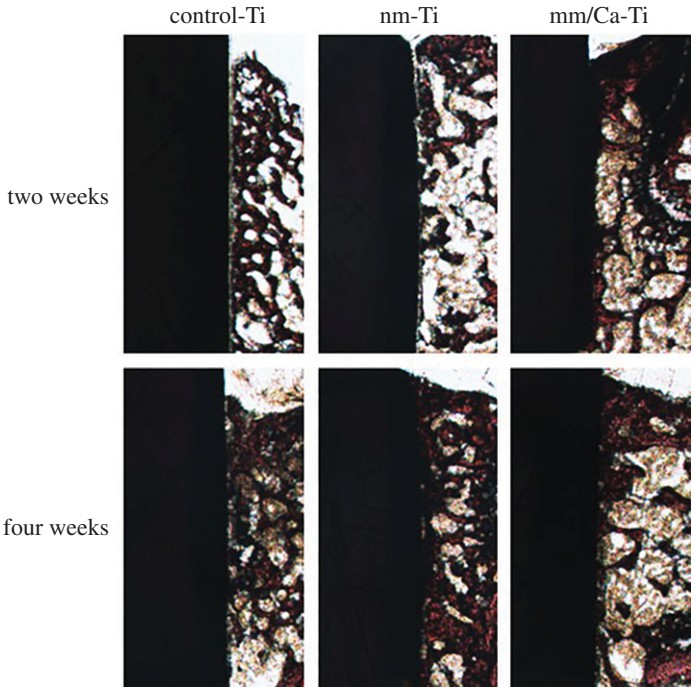

two weeks

four weeks

**Figure 9.** VG staining at two and four weeks after implantation.

control-Ti          nm-Ti          mm/Ca-Ti

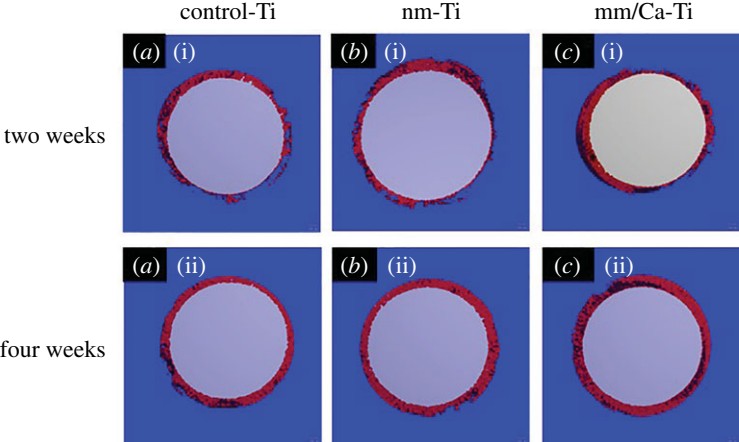

two weeks

four weeks

**Figure 10.** Micro-CT-reconstructed images at two and four weeks after implantation.

The reconstructed micro-CT images are shown in figure 10. The quantitative results are shown in figure 11. At two weeks, less bone tissue was found in the control-Ti group, while more appeared visible around nm-Ti and nm/Ca-Ti groups. nm/Ca-Ti group showed the highest values of BV/TV, Tb.N and Tb.Th. This indicated the BV in nm/Ca-Ti group was the largest since TV value of each group was equal. New bone formation of three groups showed an increasing tendency with time. After four weeks, there were no significant differences in BV/TV values among groups. Tb.Th of nm/Ca-Ti group was still the highest.

# 4. Discussion

Our previous studies showed that after hydrothermal treatment, Ca ions combined with the Ti surface, which enhanced the osteoblasts response and bone–implant contact [13,23]. In this study, we used a similar two-step hydrothermal treatment. In step 1, samples were treated in NaOH solution to create a nanoscale layer (figure 1b,e). In step 2, the same samples were hydrothermally treated in an

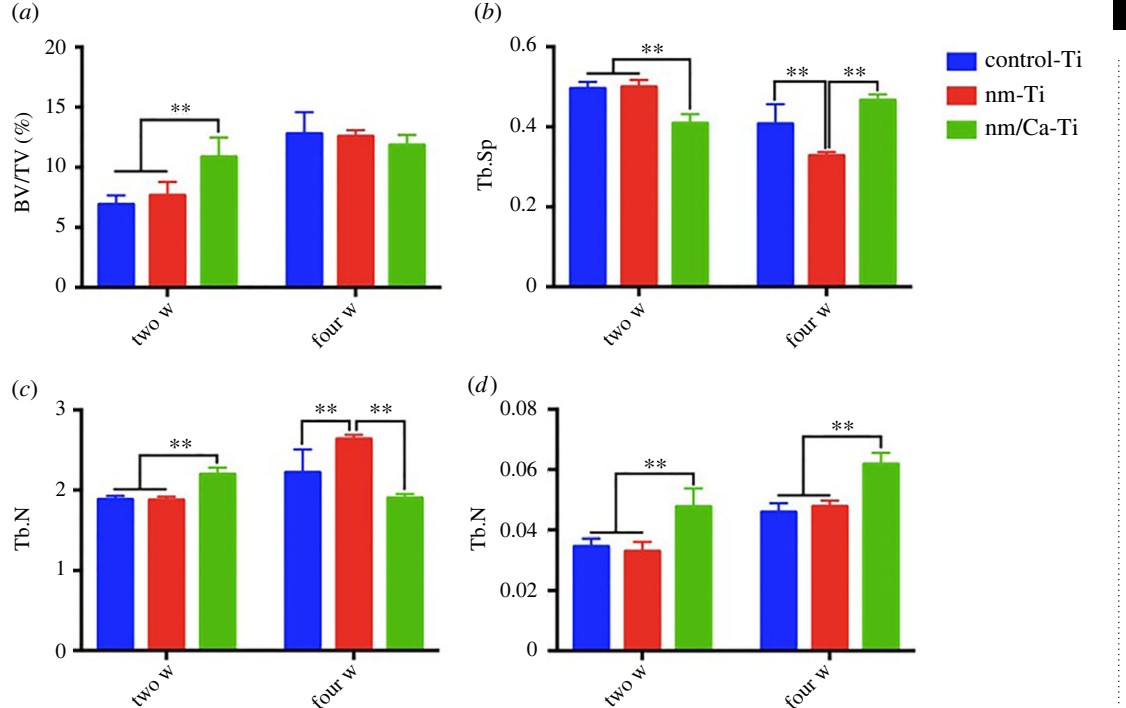

**Figure 11.** BV/TV (*a*), Tb.N (*c*), Tb.Th (*d*) and Tb.Sp (*b*) value of different implants after two and four weeks *in vivo*. (*$p < 0.05$, **$p < 0.01$).

oversaturated CaO solution. After the treatment, the nanostructure of the surface remained unchanged (figure 1*c,f*). According to table 2, Ca ions were combined with the nanosurface. At the same time, Na ions were removed. Due to the higher specific surface area of nanostructure implants, the adsorption of cell adhesion proteins was enhanced, which resulted in a better early osteoblast response [24,25]. The nanostructure could also change the osteoblast morphology to the extent of increased formation of filopodia [19]. Shah *et al.* [26] reported that, based on a larger number of implant studies, nanotextured titanium promoted bone formation and remodelling, resulting in early bone bonding to the implant surface. Our study confirmed that nanostructure promoted early osteoblasts responses including cell attachment and proliferation. After modification, implants with nanostructures possessed a lower water contact angle than before (figure 3). It is widely accepted that the hydrophilicity of the surface plays an essential role in the biological response of bone implants [14]. As shown in the cell proliferation test, the most advanced group was nm-Ti at day 1, which may be due to the highest hydrophilicity of its surface. And this result is consistent with the cell adhesion experiment (24 h).

The 7-day cell culture results showed that the nm/Ca-Ti group had the highest OD value. The surface morphology of nm-Ti group was similar to that of nm/Ca-Ti group. The most important difference between the two groups was the content of calcium ions on the surface. This indicated that hydrophilicity may not be the only factor which can affect cell proliferation, Ca ions also play an important role in this case. Therefore, loading calcium ions on the surface of nanostructure could improve the surface biological activity of titanium.

Kokubo *et al.* [22] reported that hydroxyapatite formation on a surface was a prerequisite for osseointegration. Experiments demonstrated that calcium ions in SBF can be attracted to the surface to form calcium titanate and ultimately induce the formation of apatite [27]. Therefore, SBF solution is often used for screening of bioactive materials. Based on the results of the SBF experiment (figure 4), apatite deposition was observed on both the nm-Ti and nm/Ca-Ti surfaces under SEM. However, only in the nm/Ca-Ti group was the apatite amount sufficient to be detected by XRD (figure 4*e,f*). This may be due to the formation of Ca titanate during step 2 of the hydrothermal treatment, as shown in figure 2*c*, which could act as a nucleating agent of apatite crystal growth. The results further suggested the synergistic effect of Ca ions and nanostructure.

More importantly, incorporation of Ca ions onto implant surfaces enhanced osteogenic differentiation. Differentiation of MC3T3-E1 cells on the control-Ti, nm-Ti and nm/Ca-Ti surfaces could be evaluated in terms of the mRNA expression of Runx2, BMP-2, OPN and OCN. Runx2 is

known as an osteoblast-specific transcription factor that is crucial for the differentiation of osteoblasts and can activate the expressions of downstream osteogenic genes, including OCN and OPN [27]. It can promote the differentiation of MC3T3-E1 cells into osteoblasts [28]. Liu *et al*. [29] reported that low expression of Runx2 was crucial for the maintenance of osteoblasts function. For further bone maturation, Runx2 expression had to be downregulated [30]. In the present study, Runx2 showed high-level expression at day 7 and decreased dramatically at day 10. While OCN and OPN showed opposite tendency. This result was highly consistent with the typical pattern of Runx2 expression. BMPs can promote bone formation [31]. BMP-2 is a well-known osteogenic differentiation factor that stimulates stem cell signalling pathways by activating transmembrane type I and type II receptors [32]. BMP-2 plays an important role in treating bony defects caused by surgical resection or periodontal disease [33]. OPN is commonly considered an early marker for osteogenic differentiation and plays a major role at the initial stage of biomineralization [34]. OCN is usually employed as a late-stage marker during osteoblast differentiation, and its production causes the deposition of ECM [35,36]. In this study, the expression of the early markers Runx2 and BMP-2 was highest in the nm/Ca-Ti group at day 7, which revealed that calcium ions, together with nanostructures, could accelerate osteoblast early differentiation. At day 10, late markers OCN and OPN both showed the highest level in the nm/Ca-Ti group. The results of Alizarin red staining experiment showed that calcium-loaded nanostructures significantly promoted the mineralization of extracellular matrix. The results above indicated that Ca ions, together with nanostructure, promoted the expression of early- and late-stage osteogenic differentiation markers, hence accelerate ECM mineralization.

In the animal experiment, an obvious distance was observed in the bone–implant interface in control-Ti and nm-Ti groups even after four weeks. On the contrary, in nm/Ca-Ti group, new bone was in close contact with the implant surface from two weeks. nm/Ca-Ti group also showed the highest values of BV/TV, Tb.N and Tb.Th, and the lowest value of Tb.Sp at two weeks. These results suggested that new bone formation of nm/Ca-Ti group was superior to other groups both qualitatively and quantitively. The surface of nanostructures combined with calcium ions might have promoted early osteogenesis. At four weeks, Tb.Sp of nm/Ca-Ti group was observed to increase, which probably indicated that the bone around the implant was in a more matured condition.

Taking the above results together, we can finally confirm that Ca ions showed a synergistic effect with nanostructures in the formation of early osseointegration.

The data files used in our study are available [37].

# 5. Conclusion

In this study, the surface of titanium discs was hydrothermally modified, and a calcium-bonded nanosheet with increased surface hydrophilicity was successfully fabricated. Bone-like apatite precipitation was accelerated in SBF, and the cellular responses of MC3T3-E1 cells in terms of attachment, proliferation, osteogenic differentiation and ECM mineralization were upregulated by this surface. Animal experiments confirmed that nanostructure combined with calcium could effectively promote early osteogenesis. The nanostructure and calcium ions showed synergetic effects on improving bioactivity of titanium implants. This approach may provide a new way to achieve better early osseointegration.

Ethics. The study was approved by Ethics Committee of the Affiliated Stomatological Hospital of Tongji University (2018007). All animal administrations, sample collection and procedures were performed according to the approved guidelines.

Data accessibility. The data files of the study are available at the Dryad Digital Repository: https://doi.org/10.5061/dryad.gtht76hkq [37].

Authors' contributions. Y.Z.: data curation, formal analysis, methodology and writing—original draft; J.W.: data curation, formal analysis and writing—original draft; S.H.: formal analysis and writing—review and editing; Y.Y.: formal analysis and writing—review and editing; C.Lv: data curation and formal analysis; C.Luo: data curation and formal analysis; W.Z.: data curation and formal analysis; X.S.: data curation and writing—review and editing; L.Z.: conceptualization, funding acquisition, methodology, project administration, supervision and writing—review and editing.

All authors gave final approval for publication and agreed to be held accountable for the work performed therein.

Conflict of interest declaration. We declare we have no competing interests.

Funding. We received funding from Natural Science Foundation of Shanghai (grant no. 18ZR1443000) and National Natural Science Foundation of China (grant no. 81873721).

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
