## [Peer Review File · Royal Society Open Science]

Review History

RSOS-202322.R0 (Original submission)

Review form: Reviewer 1

Is the manuscript scientifically sound in its present form?

Yes

Are the interpretations and conclusions justified by the results?

No

Is the language acceptable?

Yes

Do you have any ethical concerns with this paper?

No

Have you any concerns about statistical analyses in this paper?

No

Recommendation?

Major revision is needed (please make suggestions in comments)

Comments to the Author(s)

The authors have extended their study on hydrothermal modification of the surface of titanium discs with calcium nanosheet thereby increasing surface hydrophilicity, by showing the cellular responses of MC3T3-E1 cells. In this study, they have shown that the modification increased the attachment and proliferation, and upregulated the osteogenic differentiation of the cells. The nanostructured calcium lead to apatite formation which the authors state helped in osteoblast cell growth and differentiation. However, in my opinion this cannot be established by doing only one set of experiment, i.e showing upregulation of the osteogenic differentiation genes. The authors should see what happens as a long-term effect? Say after one month. Some basic animal experiments showing the improved /faster osseointegration need to be included for the study to be complete.

Review form: Reviewer 2

Is the manuscript scientifically sound in its present form?

No

Are the interpretations and conclusions justified by the results?

No

Is the language acceptable?

Yes

Do you have any ethical concerns with this paper?

No

Have you any concerns about statistical analyses in this paper?

No

Recommendation?

Major revision is needed (please make suggestions in comments)

Comments to the Author(s)

The manuscript entitled "Synergistic effect of nanostructure and calcium ions on improving the bioactivity of titanium implants" by Yue et al. have reported the role of hydrophilicity and nanostructure in bioactivity of titanium implants. They show that insertion of calcium ions into the titanium surface and formation of nanostructure exhibit synergetic effects on accelerating bone like apatite precipitation and osteoblast cell growth and differentiation. There are some major drawbacks of this work. A major revision is needed before consideration of this manuscript in R. Soc. open sci.

1. The author mentioned that cell adherence and proliferation is increased with increase in hydrophilicity. From water contact angle measurement as shown by the authors nm-Ti is significantly more hydrophilic than nm/Ca-Ti. So, one should expect that nm-Ti should show

better cell adherence and proliferation than nm/Ca-Ti. But the reverse is observed. The author should explain this.

2. From the surface analysis data, there is not much difference in the nanostructure between nm-Ti and nm/Ca-Ti. Then, why nm/Ca-Ti shows better activity.

3. Although the author mentioned that nm-Ti samples showed the highest cell density, from the SEM images, it looks like control-Ti and nm-Ti have much better cell density compared to nm/Ca-Ti especially at 24 hour. The author should clarify this.

4. In the gene expression study, can the author comments why the expression of Runx2 dramatically decreased in 10 days?

5. There is no mention of the concentration or the amount of control-Ti, nm-Ti and nm/Ca-Ti in any study. The author should mention about this?

6. How about checking the bioactivity with the different concentrations of calcium ions on the Ti surface to claim the importance of calcium ions in bioactivity of titanium implants.

Decision letter (RSOS-202322.R0)

Dear Dr Zhang:

Manuscript ID: RSOS-202322

Title: "Synergistic effect of nanostructure and calcium ions on improving the bioactivity of titanium implants"

Thank you for submitting the above manuscript to Royal Society Open Science. Your paper was sent to reviewers and their comments are included at the bottom of this letter.

In view of the concerns raised by the reviewers, the manuscript has been rejected in its current form. However, a new manuscript may be submitted which takes into consideration these comments.

Please note that resubmitting your manuscript does not guarantee eventual acceptance, and that your resubmission will be subject to peer review before a decision is made.

Your resubmitted manuscript should be submitted by 14-Feb-2022. If you are unable to submit by this date please contact the Editorial Office.

Yours sincerely,
Dr Laura Smith

Publishing Editor, Journals

On behalf of the Subject Editor Professor Anthony Stace and the Associate Editor Mr Andrew Dunn

REVIEWER(S) REPORTS:

Associate Editor Comments to Author ():

RSC Associate Editor:

Comments to the Author:

It is important that all points from the reviewers are addressed in full when submitting a revised version of the manuscript.

RSC Subject Editor:

Comments to the Author:

(There are no comments.)

Reviewers' Comments to Author:

Reviewer: 1

Comments to the Author(s)

The authors have extended their study on hydrothermal modification of the surface of titanium discs with calcium nanosheet thereby increasing surface hydrophilicity, by showing the cellular responses of MC3T3-E1 cells. In this study, they have shown that the modification increased the attachment and proliferation, and upregulated the osteogenic differentiation of the cells. The nanostructured calcium lead to apatite formation which the authors state helped in osteoblast cell growth and differentiation. However, in my opinion this cannot be established by doing only one set of experiment, i.e showing upregulation of the osteogenic differentiation genes. The authors should see what happens as a long-term effect? Say after one month. Some basic animal experiments showing the improved /faster osseointegration need to be included for the study to be complete.

Reviewer: 2

Comments to the Author(s)

The manuscript entitled "Synergistic effect of nanostructure and calcium ions on improving the bioactivity of titanium implants" by Yue et al. have reported the role of hydrophilicity and nanostructure in bioactivity of titanium implants. They show that insertion of calcium ions into the titanium surface and formation of nanostructure exhibit synergetic effects on accelerating bone like apatite precipitation and osteoblast cell growth and differentiation. There are some major drawbacks of this work. A major revision is needed before consideration of this manuscript in R. Soc. open sci.

1. The author mentioned that cell adherence and proliferation is increased with increase in hydrophilicity. From water contact angle measurement as shown by the authors nm-Ti is significantly more hydrophilic than nm/Ca-Ti. So, one should expect that nm-Ti should show

better cell adherence and proliferation than nm/Ca-Ti. But the reverse is observed. The author should explain this.

2. From the surface analysis data, there is not much difference in the nanostructure between nm-Ti and nm/Ca-Ti. Then, why nm/Ca-Ti shows better activity.

3. Although the author mentioned that nm-Ti samples showed the highest cell density, from the SEM images, it looks like control-Ti and nm-Ti have much better cell density compared to nm/Ca-Ti especially at 24 hour. The author should clarify this.

4. In the gene expression study, can the author comments why the expression of Runx2 dramatically decreased in 10 days?

5. There is no mention of the concentration or the amount of control-Ti, nm-Ti and nm/Ca-Ti in any study. The author should mention about this?

6. How about checking the bioactivity with the different concentrations of calcium ions on the Ti surface to claim the importance of calcium ions in bioactivity of titanium implants.

Author's Response to Decision Letter for (RSOS-202322.R0)

See Appendix A.

RSOS-220206.R0

Review form: Reviewer 2

Is the manuscript scientifically sound in its present form?

Yes

Are the interpretations and conclusions justified by the results?

Yes

Is the language acceptable?

Yes

Do you have any ethical concerns with this paper?

No

Have you any concerns about statistical analyses in this paper?

No

Recommendation?

Accept as is

Comments to the Author(s)

The quality of the manuscript is improved after the revision. It should be accepted in its' current form without further review. There is no additional comment.

Review form: Reviewer 3 (Batakrishna Jana)

Is the manuscript scientifically sound in its present form?

No

Are the interpretations and conclusions justified by the results?

No

Is the language acceptable?

Yes

Do you have any ethical concerns with this paper?

No

Have you any concerns about statistical analyses in this paper?

Yes

Recommendation?

Accept with minor revision (please list in comments)

Comments to the Author(s)

The authors provide the results from an experimental study using a multimodal approach evaluating nanostructured titanium surfaces with and without incorporated calcium ions.

1. The surface of the plates are characterised. However, the rods seem not have been characterised. I would suggest providing the characterisation of the rods since there indeed might be differences. Moreover, it would have been good to provide quantitative surface morphological data such as roughness, porosity etc if that is available.
2. The surgical procedure for installing the implants should be describe in more detail. E.g. how big were the osteotomies, how was the positions of the three different types of implants randomised across animals and positions.
3. It would be good to provide representative μ CT reconstruction images in a panel
4. The results from the animal study is unfortunately only briefly presented and discussed with the only outcome data provided is μ CT. Why have qualitative histology and histomorphometry not been performed which would have complemented the results substantially. Also, could the authors kindly give the rational for using rod shaped rather than screw shaped implants and also provide the rational for the single choice of time point. I believe that the limited data from the animal study cannot support the statement "Animal experiment further confirmed that such surface can effectively promote early osteogenesis."
5. Preferable the statistics notations in the images should be with brackets to indicate the significance more clearly
6. Update last sentence page 4 row 20 ti include "...and short term in vivo study".

Decision letter (RSOS-220206.R0)

Dear Dr Zhang:

Title: Synergistic effect of nanostructure and calcium ions on improving the bioactivity of titanium implants

Manuscript ID: RSOS-220206

The editor assigned to your paper has now received comments from reviewers. We would like you to revise your paper in accordance with the referee and Subject Editor suggestions which can be found below (not including confidential reports to the Editor). Please note this decision does not guarantee eventual acceptance.

Please submit a copy of your revised paper before 29-May-2022. Please note that the revision deadline will expire at 00.00am on this date. If we do not hear from you within this time then it will be assumed that the paper has been withdrawn. In exceptional circumstances, extensions may be possible if agreed with the Editorial Office in advance. We do not allow multiple rounds of revision so we urge you to make every effort to fully address all of the comments at this stage. If deemed necessary by the Editors, your manuscript will be sent back to one or more of the original reviewers for assessment. If the original reviewers are not available we may invite new reviewers.

Please also include the following statements alongside the other end statements. As we cannot publish your manuscript without these end statements included, if you feel that a given heading is not relevant to your paper, please nevertheless include the heading and explicitly state that it is not relevant to your work.

- Ethics statement

Please clarify whether you received ethical approval from a local ethics committee to carry out your study. If so please include details of this, including the name of the committee that gave consent in a Research Ethics section after your main text. Please also clarify whether you received informed consent for the participants to participate in the study and state this in your Research Ethics section.

OR

Please clarify whether you obtained the necessary licences and approvals from your institutional animal ethics committee before conducting your research. Please provide details of these licences and approvals in an Animal Ethics section after your main text.

OR

Please clarify whether you obtained the appropriate permissions and licences to conduct the fieldwork detailed in your study. Please provide details of these in your methods section.

- Data accessibility

It is a condition of publication that you make available the data and research materials supporting the results in the article. Datasets should be deposited in an appropriate publicly available repository and details of the associated accession number, link or DOI to the datasets must be included in the Data Accessibility section of the article (<https://royalsocietypublishing.org/rsos/for-authors#question17>). Reference(s) to datasets should also be included in the reference list of the article with DOIs (where available).

Please include a Data Availability section after your main text stating where supporting data are available from, or where they will be made available should your article be accepted for publication.

If you wish to submit your supporting data or code to Dryad (<http://datadryad.org/>), or modify your current submission to dryad, please use the following link:
<http://datadryad.org/submit?journalID=RSOS&manu=RSOS-220206>

- Competing interests

Please include a Competing Interests section after your main text declaring any financial or non-financial competing interests. If you have no competing interests please state 'I/we have no competing interests.

- Authors' contributions

Please include an Authors' Contributions section at the end of your main text detailing the contribution of each author. All authors should have read and approved the manuscript before submission and this should be stated in the Authors' Contributions section.

The list of Authors should meet all of the following criteria; 1) substantial contributions to conception and design, or acquisition of data, or analysis and interpretation of data; 2) drafting the article or revising it critically for important intellectual content; and 3) final approval of the version to be published.

- Acknowledgements

- Funding statement

Please include a funding section after your main text which lists the source of funding for each author.

Yours sincerely,
Kate Jones
Assistant Editor, Journals

RSC Associate Editor
Comments to the Author:
(There are no comments.)

Reviewers' Comments to Author:

Reviewer: 2

Comments to the Author(s)

The quality of the manuscript is improved after the revision. It should be accepted in its' current form without further review. There is no additional comment.

Reviewer: 3

Comments to the Author(s)

The authors provide the results from an experimental study using a multimodal approach evaluating nanostructured titanium surfaces with and without incorporated calcium ions.

1. The surface of the plates are characterised. However, the rods seem not have been characterised. I would suggest providing the characterisation of the rods since there indeed might be differences. Moreover, it would have been good to provide quantitative surface morphological data such as roughness, porosity etc if that is available.
2. The surgical procedure for installing the implants should be describe in more detail. E.g. how big were the osteotomies, how was the positions of the three different types of implants randomised across animals and positions.
3. It would be good to provide representative μ CT reconstruction images in a panel
4. The results from the animal study is unfortunately only briefly presented and discussed with the only outcome data provided is μ CT. Why have qualitative histology and histomorphometry not been performed which would have complemented the results substantially. Also, could the authors kindly give the rational for using rod shaped rather than screw shaped implants and also provide the rational for the single choice of time point. I believe that the limited data from the animal study cannot support the statement "Animal experiment further confirmed that such surface can effectively promote early osteogenesis."
5. Preferable the statistics notations in the images should be with brackets to indicate the significance more clearly
6. Update last sentence page 4 row 20 ti include "...and short term in vivo study".

Author's Response to Decision Letter for (RSOS-220206.R0)

See Appendix B.

RSOS-220206.R1

Review form: Reviewer 3 (Batakrishna Jana)

Is the manuscript scientifically sound in its present form?

Yes

Are the interpretations and conclusions justified by the results?

Yes

Is the language acceptable?

Yes

Do you have any ethical concerns with this paper?

No

Have you any concerns about statistical analyses in this paper?

No

Recommendation?

Accept with minor revision (please list in comments)

Comments to the Author(s)

Dear authors

Thank you very much for addressing my comments and providing the added results which in my view resulted in a stronger and very comprehensive paper.

Please review row 30-31 on page 17. "...an obvious distance was observed in the bone-implant interface even after 4 weeks". I believe that you are referring to control Ti and nm-Ti groups only (and not nm Ca-Ti) and maybe you could clarify that.

Other than that I have no further comments and consider the manuscript ready for publication.

Decision letter (RSOS-220206.R1)

Dear Dr Zhang:

Title: Synergistic effect of nanostructure and calcium ions on improving the bioactivity of titanium implants

Manuscript ID: RSOS-220206.R1

Thank you for submitting the above manuscript to Royal Society Open Science. On behalf of the Editors and the Royal Society of Chemistry, I am pleased to inform you that your manuscript will be accepted for publication in Royal Society Open Science subject to minor revision in accordance with the referee suggestions. Please find the reviewers' comments at the end of this email.

The reviewers and handling editors have recommended publication, but also suggest some minor revisions to your manuscript. Therefore, I invite you to respond to the comments and revise your manuscript.

Please also include the following statements alongside the other end statements. As we cannot publish your manuscript without these end statements included, if you feel that a given heading is not relevant to your paper, please nevertheless include the heading and explicitly state that it is not relevant to your work. We have included a screenshot example of the end statements for reference.

- Ethics statement

Please clarify whether you received ethical approval from a local ethics committee to carry out your study. If so please include details of this, including the name of the committee that gave consent in a Research Ethics section after your main text. Please also clarify whether you received informed consent for the participants to participate in the study and state this in your Research Ethics section.

OR

Please clarify whether you obtained the necessary licences and approvals from your institutional animal ethics committee before conducting your research. Please provide details of these licences and approvals in an Animal Ethics section after your main text.

OR

Please clarify whether you obtained the appropriate permissions and licences to conduct the fieldwork detailed in your study. Please provide details of these in your methods section.

- Data accessibility

It is a condition of publication that you make available the data and research materials supporting the results in the article. Datasets should be deposited in an appropriate publicly available repository and details of the associated accession number, link or DOI to the datasets must be included in the Data Accessibility section of the article (<https://royalsocietypublishing.org/rsos/for-authors#question17>). Reference(s) to datasets should also be included in the reference list of the article with DOIs (where available).

Please include a Data Availability section after your main text stating where supporting data are available from, or where they will be made available should your article be accepted for publication.

<http://datadryad.org/submit?journalID=RSOS&manu=RSOS-220206.R1>

- Competing interests

Please include a Competing Interests section after your main text declaring any financial or non-financial competing interests. If you have no competing interests please state 'I/we have no competing interests.'

- Authors' contributions

Please include an Authors' Contributions section at the end of your main text detailing the contribution of each author. All authors should have read and approved the manuscript before submission and this should be stated in the Authors' Contributions section.

The list of Authors should meet all of the following criteria; 1) substantial contributions to conception and design, or acquisition of data, or analysis and interpretation of data; 2) drafting the article or revising it critically for important intellectual content; and 3) final approval of the version to be published.

- Acknowledgements

- Funding statement

Please include a funding section after your main text which lists the source of funding for each author.

Because the schedule for publication is very tight, it is a condition of publication that you submit the revised version of your manuscript before 23-Jun-2022. Please note that the revision deadline will expire at 00.00am on this date. If you do not think you will be able to meet this date please let me know immediately.

- 1) A text file of the manuscript (tex, txt, rtf, docx or doc), references, tables (including captions) and figure captions. Do not upload a PDF as your "Main Document".
- 2) A separate electronic file of each figure (EPS or print-quality PDF preferred (either format should be produced directly from original creation package), or original software format)
- 3) Included a 100 word media summary of your paper when requested at submission. Please ensure you have entered correct contact details (email, institution and telephone) in your user account

- 4) Included the raw data to support the claims made in your paper. You can either include your data as electronic supplementary material or upload to a repository and include the relevant doi within your manuscript
- 5) All supplementary materials accompanying an accepted article will be treated as in their final form. Note that the Royal Society will neither edit nor typeset supplementary material and it will be hosted as provided. Please ensure that the supplementary material includes the paper details where possible (authors, article title, journal name).

Kind regards,
Raffaele Egizio
Assistant Editor, Journals

RSC Associate Editor:
Comments to the Author:
(There are no comments.)

RSC Subject Editor:
Comments to the Author:
This needs to go back to the previous reviewer for further assessment.

Reviewer comments to Author:
Reviewer: 3
Comments to the Author(s)
Dear authors

Thank you very much for addressing my comments and providing the added results which in my view resulted in a stronger and very comprehensive paper.

Please review row 30-31 on page 17. "...an obvious distance was observed in the bone-implant interface even after 4 weeks". I believe that you are referring to control Ti and nm-Ti groups only (and not nm Ca-Ti) and maybe you could clarify that.

Other than that I have no further comments and consider the manuscript ready for publication.

Author's Response to Decision Letter for (RSOS-220206.R1)

See Appendix C.

Decision letter (RSOS-220206.R2)

Dear Dr Zhang:

I am pleased to inform you that your manuscript entitled "Synergistic effect of nanostructure and calcium ions on improving the bioactivity of titanium implants" is now accepted for publication in Royal Society Open Science.

If you have not already done so, please ensure that you send to the editorial office an editable version of your accepted manuscript, and individual files for each figure and table included in your manuscript. This includes editable versions of Tables 1 and 2, which are currently in a picture format. You can send these in a zip folder if more convenient. Failure to provide these files may delay the processing of your proof.

Please remember to make any data sets or code libraries 'live' prior to publication, and update any links as needed when you receive a proof to check - for instance, from a private 'for review' URL to a publicly accessible 'for publication' URL. It is also good practice to add data sets, code and other digital materials to your reference list.

===COVID-SPECIFIC TEXT -- WILL ONLY BE ADDED TO COVID-PAPERS BY THE EDITORIAL OFFICE===

COVID-19 rapid publication process:

We are taking steps to expedite the publication of research relevant to the pandemic. If you wish, you can opt to have your paper published as soon as it is ready, rather than waiting for it to be published the scheduled Wednesday.

This means your paper will not be included in the weekly media round-up which the Society sends to journalists ahead of publication. However, it will still appear in the COVID-19 Publishing Collection which journalists will be directed to each week (<https://royalsocietypublishing.org/topic/special-collections/novel-coronavirus-outbreak>).

If you wish to have your paper considered for immediate publication, or to discuss further, please notify openscience_proofs@royalsociety.org and press@royalsociety.org when you respond to this email.

===END OF COVID-SPECIFIC TEXT -- WILL BE REMOVED AS NECESSARY BY THE EDITORIAL OFFICE===

Royal Society Open Science is a fully open access journal. A payment may be due before your article is published. Our partner Copyright Clearance Center's RightsLink for Scientific Communications will contact the corresponding author about your open access options from the email domain @copyright.com (if you have any queries regarding fees, please see <https://royalsocietypublishing.org/rsos/charges> or contact authorfees@royalsociety.org).

on behalf of (Associate Editor) and (Subject Editor).

Associate Editor Comments to Author:
RSC Associate Editor
Comments to the Author:
Authors have responded to all reviewer comments

RSC Subject Editor
Comments to the Author
When uploading your article for proofing, please include tables that can be edited by our proofing team

Reviewer(s)' Comments to Author:
Follow Royal Society Publishing on Twitter: @RSocPublishing
Follow Royal Society Publishing on Facebook:
<https://www.facebook.com/RoyalSocietyPublishing/>
Read Royal Society Publishing's blog:
<https://royalsociety.org/blog/blogsearchpage/?category=Publishing>

Appendix A

Our responses to the comments are as followed:

Reviewer 1:

1. The authors should see what happens as a long-term effect?

In order to evaluate the long-term effect on osteoblast differentiation, Alizarin red staining experiment was added to observe the formation of mineralized nodules of MC3T3-E1 cells on different titanium surface after osteogenesis induction for 21 days. The results showed that the amount of mineralized nodules in nm/Ca-Ti group was significantly higher than that in the other two groups. The reason we chose 21 days is that this time is commonly used in evaluation of mineralized nodules *in vitro*.

2. Some basic animal experiments showing the improved /faster osseointegration need to be included.

Animal experiments using SD rats were added. The samples were implanted into femur, retrieved at 2 weeks. The results showed that both the quantity and quality of new bone formation around the implant in the nm/Ca-Ti group was the highest. It showed that nanostructures with calcium ions loaded can significantly promote early osteogenesis *in vivo*.

Reviewer 2:

1. one should expect that nm-Ti should show better cell adherence and proliferation than nm/Ca-Ti. But the reverse is observed. The author should explain this.

Indeed, surface hydrophilicity of nm-Ti surface is the highest, and this explains its better performance in cell adhesion and early time of cell proliferation (day 1). We repeated the cell adhesion experiment and updated the SEM photographs. The characteristics of SEM pictures were consistent with the results of cell proliferation at day 1, in which nm-Ti showed higher value than the other two groups. This may due to its higher hydrophilicity. But hydrophilicity may not be the only factor which can affect cell proliferation. Ca ions are also reported to show the effect of improving cell proliferation. In the present study, from day 4, cell number in nm/Ca-Ti group increased faster than that in nm-Ti group, and became the highest at day 7. This may due to the effect of Ca loaded on nm/Ca-Ti surfaces.

2. From the surface analysis data, there is not much difference in the nanostructure between nm-Ti and nm/Ca-Ti. Then, why nm/Ca-Ti shows better activity.

The surface morphology of nm-Ti and nm/Ca-Ti was quite similar. We chose these two

surfaces expressly to minimize the influence of surface topography. The most important difference between the two surfaces was the content of calcium ions. Thus, when we find any different results of these two groups *in vitro* or *in vivo*, it will be more logical to say, that it must be the Ca, but not the surface topography which caused the differences. In the present study, nm/Ca-Ti group showed better results in bone-like apatite precipitation, osteoblast cell growth and differentiation, and promoted early osteogenesis *in vivo*. Kokubo. T. et al. reported that hydroxyapatite formation on a surface was a prerequisite for osseointegration. Therefore, Ca ions on nm/Ca-Ti may have caused early bone-like apatite precipitation *in vivo*, which finally induced faster new bone formation. Further study is needed to better reveal the whole mechanism.

3. it looks like control-Ti and nm-Ti have much better cell density compared to nm/Ca-Ti especially at 24 hours. The author should clarify this.

This maybe because the SEM pictures we chose couldn't represent the overall sample surface. We repeated the cell adhesion experiment. After checking the whole sample surface, we found that cell density in nm-Ti group was higher than the other two groups, which was consistent with day 1 results of cell proliferation experiment. SEM pictures have been updated in the manuscript.

4. In the gene expression study, can the author comments why the expression of Runx2 dramatically decreased in 10 days?

It is reported that Runx2 is an early gene in the process of osteogenesis. It can promote the differentiation of MC3T3-E1 cells into osteoblasts. Low expression of Runx2 is crucial for the maintenance of osteoblasts function. For further bone maturation, Runx2 expression has to be downregulated. In the present study, Runx2 showed high level expression at day 7, and decreased dramatically at day 10. While OCN and OPN showed opposite tendency. This result was highly consistent with the typical pattern of Runx2 expression. And the amendments have been made in the manuscript.

5. There is no mention of the concentration or the amount of control-Ti, nm-Ti and nm/Ca-Ti in any study. The author should mention about this?

In this study, control-Ti samples are commercially pure Ti without any treatment. nm-Ti samples are treated in 0.5 M sodium hydroxide solutions at 180°C for 24h. nm/Ca-Ti samples are treated in 0.5 M sodium hydroxide solutions at 180°C for 24 h, followed by treatment in 2 mM calcium oxide solutions at 180°C for 24 h. XPS analysis showed content of Ca ions on sample surfaces are: 0.22%, 0.14% and 7.21% for control-Ti, nm-

Ti and nm/Ca-Ti respectively.

6. How about checking the bioactivity with the different concentrations of calcium ions on the Ti surface to claim the importance of calcium ions in bioactivity of titanium implants.

In our early experiments, we did try different concentrations of CaO solutions (2 mM, 4 mM, 6 mM and 12 mM) for preparation of the samples. The results showed that the surface morphology was totally different. At 2 mM, the surface was nanoscale, and the calcium ion concentration is 7.21%. When the concentration reached 4 mM, 6 mM, the surface morphology changed into microscale. And the surface calcium concentration decreased at 4 mM and 6 mM, to 5.36% and 5.33% respectively. At 12 mM, although Ca content was higher (9.20%), the surface morphology was also microscale. In order to control the variables and get nanostructure surface comparable to nm-Ti, we chose the concentration of 2 mM. But still, your comment is very useful and we are planning to do further experiments to compare surfaces with different Ca content but similar topography.

Appendix B

Our responses to the comments are as followed:

Reviewer 3:

1. ...I would suggest providing the characterization of the rods since there indeed might be differences. Moreover it would have been good to provide quantitative surface morphological data such as roughness, porosity etc if that is available.

(1) In the present study, the Ti discs and rods were from the same manufacture, with similar machined surface. Therefore, we only analyzed the surface characterization of the discs and using this to represent the surface of samples used in animal study. The same logic was also seen in some other works (Cheng, M. et. al. Calcium Plasma Implanted Titanium Surface with Hierarchical Microstructure for Improving the Bone Formation. ACS Applied Materials & Interfaces. 2015, 7: 3053-13061.) But your consideration is reasonable that there might be some differences between the two surfaces, we will pay more attention to this point in future works.

(2) In our study, we mainly focused on the nanostructure of the surfaces, and lower magnification pictures showed minor differences among three groups. Surface roughness and porosity usually represents surface characterizations of micron level or even larger scale. Therefore, they were not included in the present study. But your comments are very valuable, and we will consider this in further study.

2. The surgical procedure for installing the implants should be describe in more detail. E.g. how big were the osteotomies, how was the positions of the three different types of implants randomized across animals and positions.

More details of the surgical procedure were added: A round bur with a diameter of 1.9 mm was used to prepare the implant hole with a depth of 3 mm. After the Ti cylinder was gently pushed into the hole, the muscle and skin were sutured tightly layer by layer. Three groups of titanium rods were randomly implanted into the femurs of rats, as shown in the table below.

	1	2	3	4	5	6
Left	control-Ti	control-Ti	nm-Ti	nm-Ti	nm/Ca-Ti	nm/Ca-Ti
Right	nm-Ti	nm/Ca-Ti	control-Ti	nm/Ca-Ti	control-Ti	nm-Ti

3. It would be good to provide representative μ CT reconstruction images in a panel.

Micro-CT reconstructed images were added. Data of 4 weeks after implantation were also included in

the manuscript.

The following sections were added to Materials and Methods:

3.8. Micro-CT evaluation

specimens were retrieved and fixed in 4% paraformaldehyde at 4 ° C for 48h. The specimens were examined by a micro-CT scanner (μ CT50, Scanco Medical, Switzerland). The micro-CT images were recon-structed by Evaluation Program V6.6. The bone volumes (BV) within 100 μ m around the implant were measured as trabecular bone volume fraction (BV/TV). And other quantitative analysis of bone around implants was made, including trabecular number (Tb.N), trabecular thickness (Tb.Th) and trabecular separation (Tb.Sp).

And the following paragraph in Results was updated:

The reconstructed micro-CT images were shown in Fig. 10. The quantitative results were shown in Fig. 11. At 2 weeks, less bone tissue was found in the control-Ti group, while more appeared visible around nm-Ti and nm/Ca-Ti groups. nm/Ca-Ti group showed the highest values of bone volume to total volume ratio (BV/TV), trabecular number (Tb.N) and trabecular thickness (Tb.Th). This indicated the bone volume in nm/Ca-Ti group was the largest since TV value of each group was equal. New bone formation of three groups showed an increasing tendency with time. After 4 weeks, there were no significant differences in BV/TV values among groups. Tb.Th of nm/Ca-Ti group was still the highest.

4. ...Why have qualitative histology and histomorphometry not been performed which would have complemented the results substantially. Also, could the authors kindly give the rational for using rod shaped rather than screw shaped implants and also provide the rational for the single choice of time point. I believe that the limited data from the animal study cannot support the statement “Animal experiment further confirmed that such surface can effectively promote early osteogenesis.”

(1) Qualitative histological observation at 2 and 4 weeks after implantation was added. And the results were shown in Figure 9. At 2 weeks, less new bone tissue was formed around control-Ti group. In control-Ti and nm-Ti group, an obvious distance between the bone tissue and the implant could be observed, showing a poor osseointegration. As for nm/Ca-Ti group, new bone was in close contact with the surface of the titanium implant. At 4 weeks, the surfaces with direct bone contact of all the three groups increased. But in some area of control-Ti and nm-Ti surfaces, there was still a gap between the implant and the bone.

(2) The reason we use rods rather than screws is that we tried to focus on surface structure only and avoid influences of other factors. Also because the femur of the rat is quite narrow, and the screw might increase

the difficulty of surgical procedure. Using slightly smaller round bur (1.9 mm bur compared to 2 mm rod) and careful suture can help to keep the implant stable.

(3) We agreed that the limited data from the animal study cannot support the statement “Animal experiment further confirmed that such surface can effectively promote early osteogenesis.” Therefore we changed it into: Animal experiment further indicated that such surface could promote early osteogenesis.

5. Preferable the statistics notations in the images should be with brackets to indicate the significance more clearly.

All the statistics notations were modified to better indicate the significance.

6. Update last sentence page 4 row 20 to include “...and short term in vivo study”.

“The bioactivity of the samples was investigated *in vitro*” had been updated to “The bioactivity of the samples was investigated *in vitro* and further confirmed by a short-term *in vivo* study.”

Appendix C

School and Hospital of Stomatology, Tongji University
Shanghai, China, 16/0/2022

Dear Raffaele Egizio and Royal Society Open Science editors,

Thank you very much for your E-mail of Jun. 14 regarding the resubmission of our manuscript entitled “Synergistic effect of nanostructure and calcium ions on improving the bioactivity of titanium implants” (ID: RSOS-220206.R1).

We are very pleased to know that our manuscript will be accepted for publication in your journal after minor revision. We really appreciate the editors and reviewers for all the positive and constructive comments and suggestions on our manuscript.

In accordance with the referee’s suggestion, we clarified the description of the animal experiment in discussion. The sentence was changed to “an obvious distance was observed in the bone-implant interface in control-Ti and nm-Ti groups even after 4 weeks”.

The manuscript has not been published before and is not being considered for publication elsewhere. All authors have contributed to the creation of this manuscript for important intellectual content and read and approved the final manuscript. We declare there is no conflict of interest.

Many thanks for your time and consideration.

Yours Sincerely,
Zhang Lei
(On behalf of co-authors)